# Genotype Structure Alterations in 5q SMA Patients as a Result of the Newborn Screening Program Implementation in the Russian Federation

**DOI:** 10.3390/ijms26167891

**Published:** 2025-08-15

**Authors:** Maria A. Akhkiamova, Andrey V. Marakhonov, Victoria V. Zabnenkova, Kristina A. Mikhalchuk, Sergei V. Voronin, Sergey I. Kutsev, Victoria V. Musatova, Tatyana N. Kekeeva, Nina V. Ryadninskaya, Alena L. Chukhrova, Svetlana I. Braslavskaya, Polina A. Chausova, Tatyana S. Beskorovainaya, Aleksander V. Polyakov, Kirill V. Savostyanov, Ilya S. Zhanin, Fanil S. Bilalov, Alexander L. Koroteev, Dmitry Y. Trofimov, Tatyana A. Bairova, Gulnara N. Seitova, Sergei V. Mordanov, Svetlana A. Matulevich, Elena B. Nikolaeva, Olga A. Shchagina

**Affiliations:** 1Research Centre for Medical Genetics, Moskvorechie Str., 1, 115522 Moscow, Russia; albmasha@gmail.com (M.A.A.); marakhonov@generesearch.ru (A.V.M.); zabnenkova.vv@gmail.com (V.V.Z.); grstina@yandex.ru (K.A.M.); voronin.sv@med-gen.ru (S.V.V.); kutsev@mail.ru (S.I.K.); Shkarupo@mail.ru (V.V.M.); kekeeva.genetic@gmail.com (T.N.K.); outremal@yandex.ru (N.V.R.); achukhrova@yandex.ru (A.L.C.); braslav_dna@mail.ru (S.I.B.); polinaalex85@gmail.com (P.A.C.); t-kovalevskaya@yandex.ru (T.S.B.); apol@dnalab.ru (A.V.P.); 2FSAI “National Medical Research Centre for Children’s Health”, Lomonosovskiy Prospekt, 2 b.1, 119991 Moscow, Russia; 7443333@gmail.com (K.V.S.); ilya_zhanin@outlook.com (I.S.Z.); 3SBHI Republican Medical Genetic Centre, Gafuri St., 74, 450076 Ufa, Russia; bilalov@bk.ru; 4SPB SBHI “Diagnostic Centre (Medical Genetic)”, Tobolskaya St., 5A, 194044 Saint-Petersburg, Russia; alexkoroteev@mail.ru; 5FSBI “NMRC OGP Named After Academician V.I. Kulakov”, Akademika Oparina St., 4, 117997 Moscow, Russia; molgen@bk.ru; 6FSPSI “Scientific Centre for Family Health and Human Reproduction Problems”, Timiryazeva St., 16, 664003 Irkutsk, Russia; tbairova38@mail.ru; 7FSBSI «Tomsk National Research Medical Centre of the Russian Academy of Sciences», Naberezhnaya Reki Ushaiki St., 10, 634050 Tomsk, Russia; gulnara.seitova@medgenetics.ru; 8FSBEI HE Rostov State Medical University, Nakhichevanskiy per., 29, 344022 Rostov-na-Donu, Russia; Labmed@mail.ru; 9SBHI “Krai Clinical Hospital No. 1 Named After Professor S.V. Ochapovsky”, May 1 St., 167, 350086 Krasnodar, Russia; s.matulevich@yandex.ru; 10SBHI Clinical Diagnostic Centre «Mother and Child Healthcare», Flotskaya St., 52, 620067 Ekaterinburg, Russia; eozmr-public@mis66.ru

**Keywords:** spinal muscular atrophy 5q, 5q SMA, *SMN1* exon 7 deletion, number of *SMN2* exon 7 copies, genotype, selective screening, newborn screening

## Abstract

Since 2023, the Russian Federation (RF) has implemented an expanded newborn screening (NBS) program for 36 hereditary disorders, which now includes 5q spinal muscular atrophy (5q SMA). As a result of newborn screening for 5q SMA conducted in the RF during 2023–2024, 288 newborns with a homozygous deletion of exon 7 in the *SMN1* gene were identified by molecular genetic methods. The overall observed incidence of 5q SMA was 1 in 8439 newborns, which does not significantly differ from the expected incidence of 1 in 7953 newborns, established by previous pilot screening projects (*p* > 0.05). A comparison of the genotypes of patients identified through selective and newborn screening showed statistically significant differences in the proportions of patients carrying two, three, and four or more copies of the *SMN2* gene. These findings demonstrate that the NBS program is effective in detecting both individuals with more severe phenotypes, as expected, and those with milder forms of the disease.

## 1. Introduction

Proximal spinal muscular atrophy 5q (5q SMA) is a severe autosomal recessive neuromuscular disorder characterized by progressive symptoms of muscle atrophy due to degeneration of α-motor neurons in the anterior horns of the spinal cord [1].

According to global data, the carrier frequency of the disease ranges from 1 in 40 to 1 in 60 individuals [1,2,3]; in Russia, it is estimated to be 1 in 36 individuals [4,5]. The prevalence of the disease varies by country and is estimated to be 1 case per 6000 to 10,000 live births [3,6]; the estimated incidence of 5q SMA in Russia is 1 in 5184 newborns [5].

5q SMA is caused by pathogenic variants in the *SMN1* (survival motor neuron 1) gene. A homozygous deletion of exon 7 in the *SMN1* gene accounts for the disease in approximately 95% of patients worldwide [3,6], and in 97% of patients in Russia [7].

Neonatal screening in Russia has been conducted since 1993 for the detection of two rare congenital disorders: phenylketonuria and congenital hypothyroidism. In 2006, the list was expanded to five genetic conditions: phenylketonuria, congenital hyperthyroidism, adrenogenital syndrome, cystic fibrosis, and galactosemia. In 2007, audiological screening was added to detect hearing impairment in children under the age of 1 year.

Since March 2020, the DNA Diagnostics Laboratory at the Research Centre for Medical Genetics (RCMG) has been conducting scientific and diagnostic selective screening programs for 5q SMA, aimed at confirming the diagnosis in patients with clinical symptoms of the disease.

From August 2019 to January 2022, a pilot newborn screening (NBS) program for 5q SMA was conducted in three maternity hospitals in Moscow; the observed incidence was 1 in 7801 newborns [8]. In a similar pilot screening project in Saint-Petersburg, carried out from January to November 2022, the observed incidence was 1 in 9035 newborns [9]. In 2022, a large-scale project was implemented across eight regions of the Russian Federation (Krasnodar Region, Ryazan Region, Vladimir Region, Orenburg Region, Republic of Bashkortostan, Sverdlovsk Region, Republic of North Ossetia–Alania and the Chechen Republic), involving 202,908 newborns. The incidence of 5q SMA in this cohort was determined to be 1 in 7804 [10]. Based on the aggregated data from all pilot screening projects conducted in the Russian Federation (RF), the overall observed incidence of 5q SMA was calculated to be 1 in 7953 newborns.

Here we present results of an expanded NBS program for spinal muscular atrophy 5q, which has been officially implemented in the Russian Federation since 1 January 2023 due to the possibility of early implementation of etiotropic and pathogenetic therapy.

The aim of the current study is to compare the genotypes at the SMN locus in patients with a homozygous deletion of *SMN1* exon 7 identified through newborn screening and selective screening programs.

## 2. Results

Between 6 March 2020 and 31 December 2021, a total of 2646 patients with symptoms were referred to the selective screening program for 5q SMA. As a result, the diagnosis was confirmed in 490 patients (18.5%), all of whom carried a homozygous deletion of exon 7 in the *SMN1* gene.

From 1 January 2023, to 31 December 2024, a total of 2,498,706 newborns were registered in the Russian Federation. A total of 2,430,554 of them underwent NBS for 5q SMA. The screening coverage was 98.0% in 2023 and 96.5% in 2024. During this period, 40,441 newborns were included in the risk group for hereditary disorders at 3A centers. Among them, 505 newborns comprised the risk group for 5q SMA, and their samples were sent to the 3B reference center.

As a result of NBS for 5q SMA conducted at the RCMG in 2023–2024, homozygous deletions of exon 7 in the *SMN1* gene were detected in 288 newborns (57% of the risk group).

Between 1 January 2023 and 31 December 2023, a total of 2730 patients with symptoms were referred to the newborn screening program for 5q SMA. As a result of the screening, the “5q SMA” diagnosis was confirmed in 140 patients (5.1%).

Between 1 January 2024 and 31 December 2024, the selective screening program included 2680 patients. Among them, the diagnosis of 5q SMA was confirmed in 86 cases (3.2%).

Accordingly, four comparison groups were formed: the first group comprises data from selective screening conducted in 2020–2021 (SS2020–2021); the second group includes data from newborn screening during 2023–2024 (NBS2023–2024); the third group represents selective screening data from 2023 (SS2023); and the fourth group includes data from selective screening in 2024 (SS2024) (Table 1).

We determined the genotypes of patients with a homozygous deletion of exon 7 of the SMN1 gene across all comparison groups, based on the number of *SMN2* gene copies and the presence of the *SMN1/SMN2* hybrid gene (Table 2, Figure 1).

A comparison of genotypes among patients identified through selective and newborn screening showed statistically significant differences in the proportions of patients with 2, 3, and 5 copies of the *SMN2* gene (*p* < 0.0001).

We next analyzed the patients’ age at the time of application for diagnostics in all selective screening groups. For 490 patients with 5q SMA identified through selective screening in 2020–2021 (Selective Screening Group 1, SS2020–2021), the median age was 11.4 years. The median age for 140 patients in Selective Screening Group 2 (SS2023) was 7.5 years, while for 86 patients in Selective Screening Group 3 (SS2024), the median age was 13 years.

For the purpose of analyzing the distribution of application age, the following age intervals were used: under 1 year, 1–2 years, 2–5 years, and subsequent 5-year subgroups up to 75–80 years (Table 3, Figure 2).

## 3. Discussion

During the newborn screening program in 2023–2024, the diagnosis was confirmed by molecular genetic methods for 288 newborns, which represents 57% of the primary risk group for 5q SMA.

As a result of the newborn screening program for 5q SMA conducted in the RF 2023–2024, the overall observed incidence of 5q SMA was 1 in 8439 live births (95% CI: 1 in 7692 to 1 in 9103), which did not significantly differ from the expected incidence of 1 in 7953 (95% CI: 1 in 5584 to 1 in 10,936; *p* > 0.05), as established by results from previous pilot screening projects [8,9,10].

During the course of newborn screening, two newborns with one copy of *SMN2* exon 7 were identified. No such cases were observed during selective screening in all cohorts, which can be explained by early infant mortality resulting from severe clinical manifestations of the disease. One patient with a hybrid gene and five copies of the *SMN2* gene was detected in the first selective screening group prior to the launch of the newborn screening program (Group 1, SS2020–2021). Two newborns with a hybrid gene and five *SMN2* copies were identified through newborn screening in 2023–2024. No such patients were found in the second and third selective screening groups following the implementation of the newborn screening program in 2023 and 2024, likely unnoticed due to milder clinical presentations.

Because of the non-representative sample size of patients with a single copy of the *SMN2* gene or with a hybrid gene and five *SMN2* copies, these groups were excluded from further statistical analysis.

A χ2 test with Bonferroni correction was used to compare genotypes with 2, 3, 4, and 5 copies of the *SMN2* gene, as well as genotypes including a hybrid gene and 2 to 4 copies of *SMN2*.

Statistically significant differences were found between all genotypes in the selective screening group prior to the launch of the national screening program in 2020–2021 (Group 1, SS2020–2021) and the selective screening group during the program implementation in 2023 (Group 2, SS2023) (*p* = 0.0007). The absence of differences may be attributed to the initial phase of newborn screening and the lack of substantial shifts in genotype distribution during the first year of the newborn screening program’s implementation.

Statistically significant differences were observed when comparing genotype distributions between patients from the selective screening group before the screening program (Group 1, SS2020–2021) and those from the selective screening group during 2024 (Group 3, SS2024) (*p* < 0.0001). In the 2024 selective screening group, a decrease in the proportion of patients with two *SMN2* copies and an increase in the proportions of patients with four and five copies were observed. These changes may be explained by the launch of the newborn screening program in 2023, which enabled early detection of 5q SMA patients (during the neonatal period), thereby reducing the number of such patients applying for diagnostics at later stages.

When comparing genotype distributions between the 2023 (Group 2, SS2023) and 2024 (Group 3, SS2024) selective screening groups, statistically significant differences were found (*p* = 0.0026). The manifestation of disease in patients with two copies of exon 7 of the *SMN2* gene would have been expected during these years; however, the majority of such patients were identified through newborn screening.

Statistically significant differences were observed when comparing genotype distributions between the selective screening group prior to the start of the national screening program (Group 1, SS2020–2021,) and the newborn screening group from 2023 to 2024 (*p* < 0.0001).

In the 2023–2024 newborn screening group, an increased proportion of patients with 2, 4, and 5 copies of the *SMN2* gene and a decreased proportion of patients with 3 copies were noted. These differences likely reflect the true distribution of SMN locus genotypes in the newborn screening population, unaffected by age-related onset of clinical symptoms.

Statistically significant differences were also found when comparing the 2023–2024 newborn screening group with both the selective screening group from 2023 (Group 2, SS2023) (*p* < 0.0001) and the selective screening group from 2024 (Group 3, SS2024) (*p* < 0.0001). In the 2023 selective screening group, a decreased proportion of patients with two and five *SMN2* copies and an increased proportion of patients with three copies were observed, compared to the 2023–2024 newborn screening group.

In the 2024 selective screening group, a decreased proportion of patients with two copies and increased proportions with three and five copies were observed, compared to the 2023–2024 newborn screening group.

These differences are attributable to the full spectrum of genotypes detected through newborn screening, the reduced proportions of genotypes with two copies within the selective screening groups, and the predominance of clinically milder forms of 5q SMA detected through selective screening. Taking into account the Bonferroni correction, the significance level was 0.0083 (Figure 3).

As a result of comparing the genotypes in the 2023–2024 newborn screening group with those of all selective screening groups, statistically significant differences were registered for the genotype with two copies of exon 7 of the SMN2 gene (*p* < 0.0001). The proportion of patients with two copies of the *SMN2* gene was significantly higher in the newborn screening group compared to all selective screening groups. These differences may be explained by the effective identification of the most severe cases through newborn screening, as well as by early infant mortality due to severe clinical manifestations, which likely led to the decrease in numbers of such patients in selective screening programs.

Comparison of genotypes between the newborn screening group 2023–2024 and all other selective screening groups revealed statistically significant differences with respect to the genotype with two copies of *SMN2* exon 7 (*p* < 0.0001). The proportion of patients with two copies of *SMN2* was significantly higher in the newborn screening group compared to the other selective screening groups. These differences may be related to superior detection of severe cases in newborn screening, as well as early infant mortality in this group due to severe clinical manifestations and failure to diagnose such patients in selective screening programs.

Statistically significant differences were also observed when comparing the 2023–2024 neonatal screening group with the first (Group 1, SS2020–2021) and second (Group 2, SS2023) selective screening groups for the genotype with three copies of exon 7 of the *SMN2* gene (*p* < 0.0001). The proportion of patients with three copies of *SMN2* was lower in the newborn screening group, which may reflect the prevalence of more severely affected individuals with two copies identified through newborn screening.

Comparing the NBS2023–2024 group with the SS2020–2021 and SS2023 groups for patients with the genotype carrying three copies of *SMN2* exon 7 (*p* < 0.0001), the proportion of patients with three copies of *SMN2* was lower in the neonatal screening group, which is explained by the higher representation of more severely affected patients carrying two copies identified during neonatal screening.

Among all the groups of selective screening and newborn screening, statistically significant differences were found in the genotype with four copies of exon 7 of the *SMN2* gene (*p* = 0.0004), due to a decrease in the proportion of patients in Group 1, SS2020–2021.

Statistically significant differences were detected when comparing the first selective screening group (Group 1, SS2020–2021) with both the third selective screening group (Group 3, SS2024) and the newborn screening group (NBS2023–2024) for the genotype with five copies of exon 7 of the *SMN2* gene (*p* < 0.0001). In the 2024 selective screening group, an increased proportion of patients with five copies was observed. This may be explained by a higher rate of individuals with a “mild” phenotype seeking medical evaluation due to increased public awareness regarding 5q SMA following the implementation of neonatal screening.

No statistically significant differences were found across all pairwise comparisons between the three selective screening groups and the newborn screening group for hybrid genotypes with two copies (*p* = 0.4228) and three copies (*p* = 0.0984) of exon 7 of the *SMN2* gene, except for four copies (*p* = 0.0025). The absence of significant differences in hybrid genotypes likely reflects their rarity in the Russian population.

All pairwise comparisons were performed using the Bonferroni correction equal to 0.0083 (Figure 4). The observed statistical differences across genotypes demonstrate a shift in the structure of SMN locus genotypes among patients with symptoms following the implementation of large-scale neonatal screening in the RF.

When comparing a cohort of newborns with homozygous deletion of exon 7 in the *SMN1* gene identified through newborn screening in 2023–2024 to patients with 5q SMA identified through selective screening prior to the launch of pilot newborn screening programs in the Russian Federation (Group 1, SS2020–2021) and to those who applied during the newborn screening period in 2023 and 2024, statistically significant differences in the distribution of SMN locus genotypes were observed.

Prior to the screening implementation, the majority of patients had three copies of the *SMN2* gene, while genotypes with two or four copies of *SMN2* were less common. The introduction of newborn screening radically altered this distribution, thus showing the true proportions of SMN locus genotypes. Specifically, the proportion of newborns with two copies of *SMN2* (typically associated with a severe disease course) increased in comparison with those in the selective screening groups. Conversely, the proportion of newborns with three copies of *SMN2* decreased relative to those groups. We also noted an increase in the proportion of newborns with four and five copies of *SMN2*, as well as the diversity of hybrid *SMN1/SMN2* genes with varying copy numbers of *SMN2*, all of which are generally associated with milder disease phenotypes.

Thus, when comparing SMN locus genotypes detected before the screening implementation to those detected during the newborn screening period, we can assume that patients with “severe” genotypes were previously undiagnosed due to early childhood mortality, while those with “milder” genotypes were often overlooked by the medical community. In the selective screening groups of 2023 and 2024, a reduction in the proportion of patients with two copies of *SMN2* was observed, which can be explained by the identification of such genotypes through newborn screening. Additionally, an increase in the proportion of patients with four and five copies of *SMN2* was noted in the selective screening cohorts of 2023 and 2024. These differences are caused by the early detection of newborns with 5q SMA through newborn screening, leading to a shift in the genotype distribution within the selective screening cohorts during the screening period in favor of “milder” genotypes.

The experience of other countries in implementing newborn screening for 5q SMA is presented in a global data review conducted by a group of researchers. A survey form was distributed to experts in 152 countries, and responses were received from 82 countries (54%). The research was conducted between 26 November and 29 December, 2020, and the results were published in a 2021 article. In total, these 82 countries account for 8,434,000 live births annually, representing 57% of the global annual birth rate. Newborn screening for 5q SMA had been implemented in nine of the respondent countries, constituting 11% of all participants. The data presented cover the period from 2014 to 2020. In total, respondents from these nine countries reported 288 newborns diagnosed with 5q SMA out of 3,674,277 newborns who underwent respective national screening programs. The reported disease prevalence was 1 in 12,757 newborns [11].

This study did not examine the genotype spectrum of newborns with 5q SMA. However, a comparison of the disease prevalence in the nine reporting countries with that observed in the Russian Federation allows to suggest higher 5q SMA prevalence in Russia. This may indicate a more rigorous and comprehensive approach to the identification of newborns with 5q SMA in the Russian Federation.

The implementation of newborn screening in various countries is a necessary diagnostic measure. Due to the availability of therapy, detection and early treatment of 5q SMA in newborns is the main goal of screening programs worldwide.

## 4. Materials and Methods

### 4.1. Selective 5q SMA Screening Program Design

Patients referred to 5q SMA diagnostics as a result of selective screening met various criteria for the diagnosis of 5q SMA. The main characteristic symptoms of the disease include decreased or absent pharyngeal and glossal reflexes, impaired breathing, scoliosis [12], symmetrical hypotrophy and hypotonia of proximal muscles in lower and upper limbs [13,14], symmetrical decrease or absence of deep tendon reflexes in limbs, muscle fasciculations [15], postural hand tremor, positive Gowers’ sign, and waddling gait [16].

Selective screening programs for 5q SMA have been conducted at the Laboratory DNA diagnostics, RCMG, for the past five years for symptomatic patients. This study includes data from the following groups:Group 1 (Selective Screening Prior to Nationwide Screening Projects, SS2020–2021): A total of 2646 patients who underwent selective screening for 5q SMA before the launch of pilot screening and newborn screening projects, from 6 March 2020, to 31 December 2021.Group 2 (Selective Screening During NBS in 2023, SS2023): A total of 2730 patients referred to the RCMG during the implementation of the NBS program from 1 January 2023, to 31 December 2023.Group 3 (Selective Screening During NBS in 2024, SS2024): A total of 2680 patients referred to the RCMG during the implementation of the NBS program from 1 January 2024, to 31 December 2024.

Patients with clinical symptoms referred to the RCMG in 2022 (SS2022) for 5q SMA diagnostics were not included in the study due to the large-scale pilot screening projects for 5q SMA conducted in the RF during that year (Figure 5).

### 4.2. Nationwide-Expanded Newborn Screening Program Design

Since 1 January 2023, newborn screening has been conducted in all 89 federal subjects of the Russian Federation for congenital metabolism disorders, primary immunodeficiencies, and 5q SMA.

Blood samples are collected as three blood spots on filter paper cards between 24 and 48 h of life for full-term newborns, and between 144 and 168 h for preterm newborns. First-tier screening is conducted at one of the ten 3A regional centers and performed using real-time PCR with the TK-SMA (Generium, Russia) and NeoScreen SMA/TREC/KREC (DNA-Technology, Russia) kits. The goal of the first stage of screening is mass newborn screening to form a risk group. Following the first-tier stage, newborns with positive or doubtful results undergo confirmatory testing using whole blood samples collected in tubes with EDTA. These samples are then sent to a single 3B reference center: the Research Centre for Medical Genetics (RCMG).

In accordance with the Order of the Ministry of Healthcare of the Russian Federation No. 274n dated 21 April 2022, “On the approval of the procedure for providing medical care to patients with congenital and/or hereditary diseases,” the 3B reference center for SMA confirmatory diagnostics is the Research Centre for Medical Genetics [17].

At the RCMG, confirmatory diagnostics for 5q SMA are carried out, including detection of the homozygous deletion of exon 7 in the *SMN1* gene and determination of the *SMN2* exons 7 and 8 copy number. The RCMG reference center is the final instance in molecular diagnostics for 5q SMA [17,18]. Newborn screening includes mass newborn examination for congenital and/or hereditary diseases in order to preemptively detect the disorders and initiate timely treatment to prevent early death and disability in childhood. The implementation of therapy is decided by a Concilium in regional centers.

This study examines the results of newborn screening for 5q SMA in a cohort of 2,430,554 newborns tested between 1 January 2023 and 31 December 2024.

DNA diagnosis

Genomic DNA was extracted from whole blood using the Wizard^®^ Genomic DNA Purification Kit (Promega, Madison, WI, USA) according to the manufacturer’s protocol. To detect the homozygous deletion of exon 7 in the *SMN1* gene, the Melt Assay SALSA^®^ MC002 SMA Newborn Screen kit (MRC Holland, Amsterdam, The Netherlands) was used in accordance with the manufacturer’s protocol. Reaction products were analyzed by melting curve analysis using the Applied Biosystems™ QuantStudio™ 5 Real-Time PCR system (Thermo Fisher Scientific, Waltham, MA, USA). Real-time PCR data was processed using the Design & Analysis 2.6.0 (DA2.6.0) software (Thermo Fisher Scientific, Waltham, MA, USA).

To determine the copy number of exons 7 and 8 of the *SMN1* and *SMN2* genes, the SALSA^®^ MLPA^®^ Probemix P060 SMA Carrier and SALSA^®^ MLPA^®^ Probemix P021 SMA kits (MRC Holland, Amsterdam, The Netherlands) were used in parallel, according to the manufacturer’s protocol, with subsequent fragment analysis on the ABI Prism^®^ 3500 Genetic Analyzer (Thermo Fisher Scientific, Waltham, MA, USA).

A homozygous deletion of exons 7 and 8 in the *SMN1* gene is the primary cause of 5q SMA. However, there have been cases with hybrid *SMN1/SMN2* genes in a compound heterozygous state with a deletion of exons 7 and 8 in SMN1 contributing to the development of 5q SMA. The formation of these SMN1/SMN2 hybrid genes is most likely the result of Alu-mediated non-allelic homologous recombination events within the SMN locus, intrachromosomal deletion followed by fusion of the 5′ end of *SMN2* with the 3′ end of *SMN1*, or partial *SMN1* to *SMN2* gene conversion, in which the flanking regions of exon 8 of *SMN1* and exon 7 of *SMN2* are merged [19,20].

The MLPA data was processed using the Coffalyser v.8 software, taking into account the quality control of the test and control samples in accordance with the manual provided by the manufacturer of the MLPA kit (MRC Holland, Amsterdam, the Netherlands).

Statistical analysis was carried out using the χ2 test with Bonferroni correction and Spearman’s rank correlation coefficient.

## 5. Conclusions

5q spinal muscular atrophy is one of the most common orphan diseases worldwide. The implementation of newborn screening across various countries is a necessary measure for the diagnosis of such patients. Given the availability of etiological gene therapy, the identification and early treatment of newborns with 5q SMA is a primary objective of screening programs globally. In the RF, both etiotropic gene and pathogenetic therapy are actively administered not only to newborns but also to patients with symptoms. As a result, the traditional classification of 5q SMA types is becoming less relevant due to the introduction of newborn screening and the initiation of therapy at the presymptomatic stage of the disease.

A comparative analysis of genotypes from 490 patients identified through selective 5q SMA screening prior to the launch of pilot projects and newborn screening during 2020–2021, 288 newborns identified through neonatal screening in 2023–2024, and an additional 140 and 86 patients identified through selective screening during the newborn screening period in 2023 and 2024, respectively, showed statistically significant differences in the proportions of patients carrying two, three, and four or more copies of the *SMN2* gene. Newborn screening enabled the identification of newborns with 5q SMA during the first year of life. Aside from that, newborn screening effectively detects not only newborns with the most “severe” genotypes but also those with “milder” ones.

Thus, the implementation of newborn screening has significantly altered the genotype distribution among 5q SMA patients with symptoms and has made it possible to determine the true structure of the disease.

## Figures and Tables

**Figure 1 ijms-26-07891-f001:**
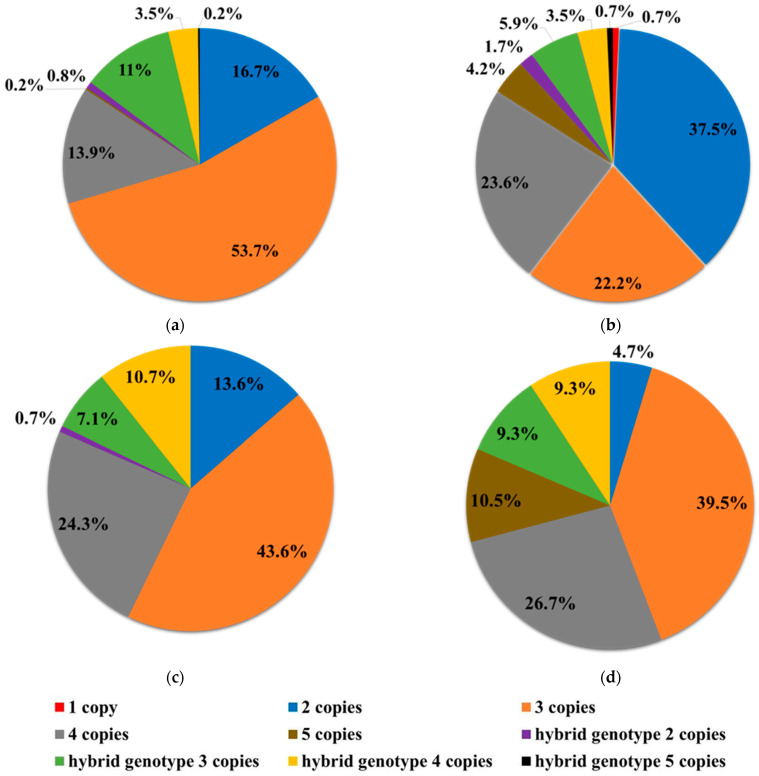
Distribution of *SMN2* exon 7 copy numbers: (**a**) Distribution of *SMN2* exon 7 copy numbers among 490 patients with a homozygous deletion of *SMN1* exon 7, identified through selective 5q SMA screening prior to the launch of pilot projects and newborn screening in 2020–2021 (Selective Screening Group 1, SS2020–2021); (**b**) distribution of *SMN2* exon 7 copy numbers among 288 newborns with a homozygous deletion of SMN1 exon 7, identified through newborn screening in 2023–2024 (NBS2023–2024); (**c**) distribution of *SMN2* exon 7 copy numbers among 140 patients with a homozygous deletion of *SMN1* exon 7, identified through selective 5q SMA screening during the implementation of newborn screening in 2023 (Selective Screening Group 2, SS2023); (**d**) distribution of *SMN2* exon 7 copy numbers among 86 patients with a homozygous deletion of *SMN1* exon 7, identified through selective 5q SMA screening during the implementation of newborn screening in 2024 (Selective Screening Group 3, SS2024).

**Figure 2 ijms-26-07891-f002:**
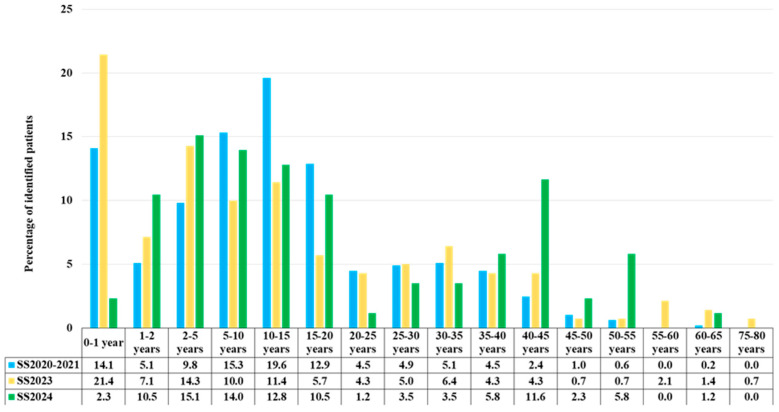
Histogram of the age distribution at the time of application among 490 patients in 2020–2021 (Group 1, SS2020–2021), 140 patients screened in 2023 (Group 2, SS2023), and 86 patients in 2024 (Group 3, SS2024), shown as percentages, all with a confirmed diagnosis of 5q SMA through the selective screening program.

**Figure 3 ijms-26-07891-f003:**
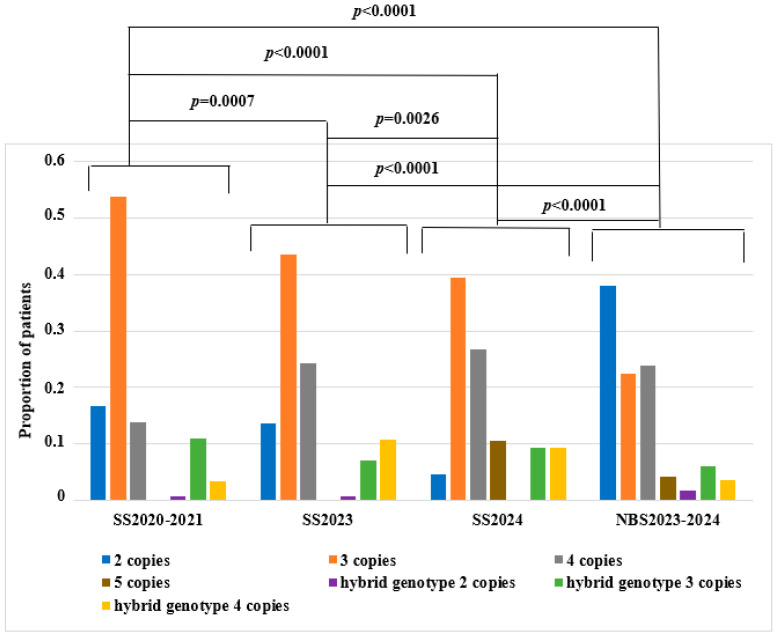
Histogram of genotype distribution across all selective screening (SS) groups and newborn screening (NBS) groups by year, along with the results of pairwise statistical comparisons using the χ2 test with Bonferroni correction.

**Figure 4 ijms-26-07891-f004:**
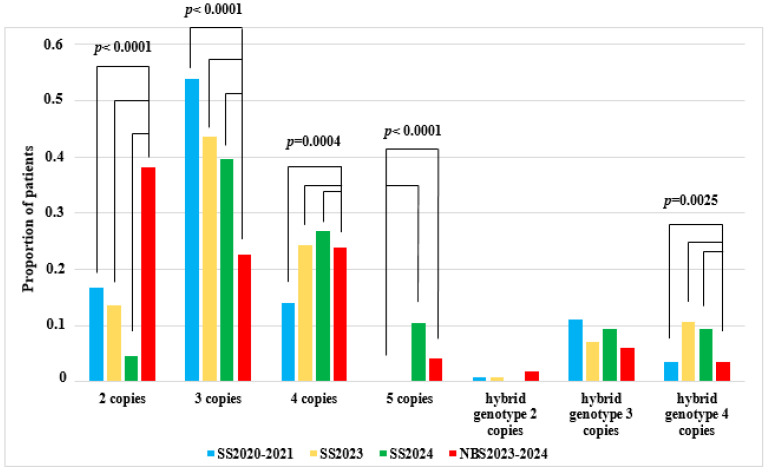
Histogram of genotype proportions across all three selective screening (SS) groups and the newborn screening (NBS) group, categorized by the number of *SMN2* gene copies and the presence of a hybrid gene.

**Figure 5 ijms-26-07891-f005:**
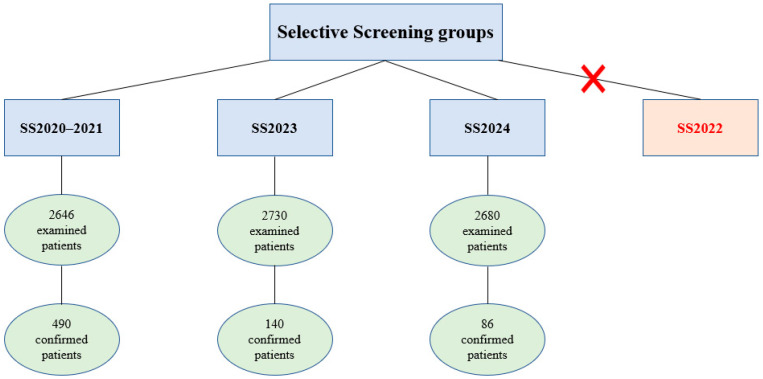
The block diagram of the three selective screening groups included in the study: Selective Screening Prior to Nationwide Screening Projects, SS2020–2021, Selective Screening During NBS in 2023, SS2023, Selective Screening During NBS in 2024, SS2024; and the selective screening group excluded from the study—SS2022.

**Table 1 ijms-26-07891-t001:** Number of analyzed samples and detected homozygous deletions of exon 7 of the *SMN1* gene in three selective screening (SS) groups and the newborn screening (NBS) group, shown in absolute numbers and percentages.

Samples	SS2020–2021		NBS2023–2024		SS2023		SS2024	
	number	%	number	%	number	%	number	%
Samples analyzed	2646	100	505	100	2730	100	2680	100
Deletions detected	490	18.5	288	57	140	5.1	86	3.2

**Table 2 ijms-26-07891-t002:** Genotype spectrum of the SMN locus in three selective screening (SS) groups and the newborn screening (NBS) group, shown in absolute numbers and percentages.

Genotype	SS2020–2021		NBS2023–2024		SS2023		SS2024	
	number	%	number	%	number	%	number	%
1 *SMN2* copy	0	0	2	0.7	0	0	0	0
2 *SMN2* copies	82	16.7	108	37.5	19	13.6	4	4.7
3 *SMN2* copies	263	53.7	64	22.2	61	43.6	34	39.5
4 *SMN2* copies	68	13.9	68	23.6	34	24.3	23	26.7
5 *SMN2* copies	1	0.2	12	4.2	0	0	9	10.5
hybrid gene 2 *SMN2* copies	4	0.8	5	1.7	1	0.7	0	0
hybrid gene 3 *SMN2* copies	54	11	17	5.9	10	7.1	8	9.3
hybrid gene 4 *SMN2* copies	17	3.5	10	3.5	15	10.7	8	9.3
hybrid gene 5 *SMN2* copies	1	0.2	2	0.7	0	0	0	0
Total	490	100	288	100	140	100	86	100

**Table 3 ijms-26-07891-t003:** Distribution of age at the time of application among 490 patients in 2020–2021 (Group 1, SS2020–2021), 140 patients screened in 2023 (Group 2, SS2023), and 86 patients in 2024 (Group 3, SS2024), shown in absolute numbers and percentages, all with a confirmed diagnosis of 5q SMA through the selective screening program.

Age, Years	SS2020–2021		SS2023		SS2024	
	number	%	number	%	number	%
0–1	69	14.1	30	21.4	2	2.3
1–2	25	5.1	10	7.1	9	10.5
2–5	48	9.8	20	14.3	13	15.1
5–10	75	15.3	14	10.0	12	14.0
10–15	96	19.6	16	11.4	11	12.8
15–20	63	12.9	8	5.7	9	10.5
20–25	22	4.5	6	4.3	1	1.2
25–30	24	4.9	7	5.0	3	3.5
30–35	25	5.1	9	6.4	3	3.5
35–40	22	4.5	6	4.3	5	5.8
40–45	12	2.4	6	4.3	10	11.6
45–50	5	1.0	1	0.7	2	2.3
50–55	3	0.6	1	0.7	5	5.8
55–60	0	0	3	2.1	0	0
60–65	1	0.2	2	1.4	1	1.2
75–80	0	0	1	0.7	0	0
Total	490	100	140	100	86	100

## Data Availability

Original materials involved in the study are included into the article; additional requests may be sent to the corresponding author.

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
