# Peer review of "Genotype Structure Alterations in 5q SMA Patients as a Result of the Newborn Screening Program Implementation in the Russian Federation"

_ijms, 2025, doi:10.3390/ijms26167891_

Round 1
Reviewer 1 Report
Comments and Suggestions for Authors
International Journal of Molecular Sciences
Dear Dr, Maria A Akhkiamova
I am honored to be involved in reviewing the manuscript entitled: Genotype structure alterations in 5q SMA patients as a result of the newborn screening program implementation in the Russian Federation, Manuscript ID: ijms-3776304. This study is very interesting. This study aims to compare the genotypes at the SMN locus in patients with a homozygous deletion of SMN1 exon 7 identified through newborn screening and selective screening programs. The conclusion draw from the article is that the newborn screening (NBS) program is effective in detecting both individuals with more severe phenotypes, as expected, and those with milder forms of the disease. However, there are still many problems in the article, so it is not recommended to publish it. The specific problems are as follows:
1. Materials and Methods
“4. Materials and Methods” should be moved to the front of "3. Results " section。Suggest describing the lines in the following order
1.1 Subjects include the inclusion and exclusion of research subjects, and it is best to add a flowchart to show the entire process of case inclusion and exclusion. It is advisable to add diagnostic criteria for SMA and cite relevant literature.
1.2 Instruments and reagents
1.3 Specimen Collection, Testing, and Quality Control
1.4 Screening plan design
1.5 Statistical analysis
2. Results
2.1 The row and column contents of Tables 1, 2, and 3 should be replaced with each other, referring to the representation format in Figure 2. I suggest you present categorical variables as N (%).
2.2 You should add a “total” column in the Tables 2 and 3.
2.3 The expression of P value of pairwise comparison in each group is too chaotic in Fig. 3. It is suggested to simplify Fig. 3 and list only the most important inter group P values.
3. Discussion
The discussion section mostly describes screening methods. It is recommended to rewrite the discussion section. The following order is for reference.
3.1 Main findings
3.2 Provide a comprehensive discussion of the results obtained from this study to demonstrate the reliability and limitations of your findings, and cite relevant literature to support or refute your argument
3.3 Clinical significance of the study
3.4 Limitations of the study
3.5 Conclusions
4. References
The number of references is relatively small, and more references should be added.
Author Response
Comment 1: [ Materials and Methods.“4. Materials and Methods” should be moved to the front of "3. Results " section]
Response 1 : [Dear Reviewer 1, thank you so much for your edits. All editorial changes are highlighted in yellow. Dear reviewer 1, unfortunately, the rules of the journal do not provide for the transfer of the section.]
Suggest describing the lines in the following order
Comment 1.1 [Subjects include the inclusion and exclusion of research subjects, and it is best to add a flowchart to show the entire process of case inclusion and exclusion. It is advisable to add diagnostic criteria for SMA and cite relevant literature.]
Response 1.1: [The block-diagram illustrating the entire process of including and excluding cases is presented in the corresponding section. Diagnostic criteria for SMA 5q and references to the relevant literature are presented in the corresponding section.]
Comment 1.2: [Instruments and reagents.]
Response 1.2: [Dear reviewer, this section is described in detail.]
Comment 1.3: [Specimen Collection, Testing, and Quality Control.]
Response 1.3: [The "Sample collection" and "Testing" sections are listed in the appropriate order, and quality control has also been added.]
Comment 1.4: [Screening plan design.]
Response 1.4: [The objectives of the study did not include the development of a screening plan. The screening plan is described in detail in order 274n in 11 on the list of references.]
Comment 1.5: [Statistical analysis.]
Response 1.5: [Statistical analysis is described last.]
- Results
Comment 2.1: [The row and column contents of Tables 1, 2, and 3 should be replaced with each other, referring to the representation format in Figure 2. I suggest you present categorical variables as N (%).]
Response 2.1: [Dear reviewer, unfortunately, we cannot replace the contents of rows and columns of Tables 1, 2 and 3 with each other, referring to the presentation format in Figure 2. Because the tables display the completeness of the data in both percentages and absolute values. Only Table 3 is relevant to Figure 2. Table 1 reflects the samples. Table 2 shows the genotypes.]
Comment 2.2: [You should add a “total” column in the Tables 2 and 3.]
Response 2.2: [The total section has been added.]
Comment 2.3: [The expression of P value of pairwise comparison in each group is too chaotic in Fig. 3. It is suggested to simplify Fig. 3 and list only the most important inter group P values.]
Response 2.3: [Dear reviewer, unfortunately, Figure 3 cannot be simplified because it contains all the most important intergroup P values.]
Comment 3: [Discussion.The discussion section mostly describes screening methods. It is recommended to rewrite the discussion section. The following order is for reference.]
Response 3: [Dear reviewer, the screening methods are not described in the discussion section. This section contains a comparison of groups by genotype with statistical data, as well as an explanation of these differences or their absence.]
Comment 3.1: [Main findings.]
Response 3.1: [The main conclusions are listed in the conclusions section, and the discussion section describes each comparison in detail in order.]
Comment 3.2 Provide a comprehensive discussion of the results obtained from this study to demonstrate the reliability and limitations of your findings, and cite relevant literature to support or refute your argument
Response 3.2 [A comprehensive discussion of the results is provided, but unfortunately it is not possible to provide links to the relevant literature on genotype comparison. The authors describe pilot projects for screening for SMA, but there are not enough publications on the screening itself, especially since there are no articles comparing the genotypes of neonatal and selective screenings in other countries. One major collective article is presented in my work.]
Comment 3.3: [Clinical significance of the study.]
Response 3.3: [The clinical significance of the study has been added to the relevant section.]
Comment 3.4: [Limitations of the study/]
Response 3.4: [Response The limitations of the study are presented in the materials and methods section, as well as during the discussion of comparing groups by genotype to better understand why a particular group did not pass statistical analysis.]
Comment 3.5: [Conclusions.]
Response 3.5: [The conclusions of the discussion are presented taking into account the comparison with 1 global article and are supported by clinical significance.]
Comment 4: [References. The number of references is relatively small, and more references should be added.]
Response 4: [Dear reviewer, the list of references has been expanded.]
Reviewer 2 Report
Comments and Suggestions for Authors
The article is interesting and generally well-written. Regarding the description of the gene names SMN1 and SMN2, I suggest writing them in uppercase letters and italicized to follow the standard nomenclature in the field. I think the article lacks an explanation related, for example, to what a hybrid gene would be in the context of SMN and the relationship between the number of copies of the SMN2 gene with clinical presentation/phenotype. This could be addressed within the discussion when referencing these findings. I believe this would also help enrich the article further and clarify, from a molecular perspective, the implications of these findings. I am not sure if this is a standard of the journal, but I believe if the methodology came before the results, the reading of the article would be better and more linear, which would facilitate its better understanding.
Author Response
Comment: [The article is interesting and generally well-written. Regarding the description of the gene names SMN1 and SMN2, I suggest writing them in uppercase letters and italicized to follow the standard nomenclature in the field. I think the article lacks an explanation related, for example, to what a hybrid gene would be in the context of SMN and the relationship between the number of copies of the SMN2 gene with clinical presentation/phenotype. This could be addressed within the discussion when referencing these findings. I believe this would also help enrich the article further and clarify, from a molecular perspective, the implications of these findings. I am not sure if this is a standard of the journal, but I believe if the methodology came before the results, the reading of the article would be better and more linear, which would facilitate its better understanding.]
Response: [Dear Reviewer 1, thank you so much for your edits. All editorial changes are highlighted in blue. Dear reviewer, thank you for your comments. The name of the genes has been corrected to italics everywhere. The description of the hybrid gene in the article is presented, highlighted in blue. Thank you very much for the suggestion, but describing the genotype/phenotype picture relative to the number of copies of copies of standard and hybrid genotypes is not part of the main task of the study. I have published an article on this topic in terms of a literary review on disease severity modifiers. In addition, we do not provide data on standard genotypes relative to the clinical picture, therefore, it is not advisable to provide literature data on hybrid genotypes.Dear reviewer, I absolutely agree with you. But unfortunately, the rules of the journal do not make it possible to place the materials and methods section before the results.]
Reviewer 3 Report
Comments and Suggestions for Authors
The manuscript presents a valuable and timely evaluation of the implementation of nationwide newborn screening (NBS) for 5q spinal muscular atrophy (SMA) in the Russian Federation. The authors analyse outcomes based on over 2.4 million screened newborns and an additional cohort identified through selective screening, with a focus on SMN2 copy number distribution. The study addresses an important clinical and public health issue, particularly relevant in the context of early therapeutic interventions for SMA.
The manuscript has clear strengths, including the large cohort, the focus on real-world NBS implementation, and the relevance to other countries scaling up similar programs.
However, certain sections would benefit from clarification, additional context, and minor restructuring to enhance readability and scientific clarity for an international audience.
- Study Aim:
- The aim of the study should be explicitly stated, preferably at the end of the Introduction section.
- NBS Policy Background:
- Consider adding a dedicated paragraph in the Introduction or Discussion summarising the expansion of Russia's NBS program in 2023 (to include SMA and other conditions) and explaining how SMA was added to the national panel.
- Discuss the public health significance of early SMA detection within the Russian healthcare context.
- Terminology – Selective vs. Universal Screening:
- The phrase “selective screening during NBS” (e.g., lines 350, 353) may be misleading, as NBS is defined as universal screening shortly after birth. Please clarify whether these patients were truly part of the NBS program or were referred for testing due to symptoms or other risk factors.
- You may wish to add a clear definition of “selective screening” vs. “universal NBS” early in the manuscript (e.g., Methods or Introduction).
- Reference to National Policy and Order 274n:
- When referring to Order 274n (line 371), briefly explain its relevance for confirmatory testing and follow-up. It may be helpful to cite the document and explain how it mandates centralised procedures.
- Center Classification (3A and 3B):
- Clarify the roles of reference centers categorised as 3A and 3B (lines 364–368). Define their responsibilities in the NBS-confirmation-treatment workflow and consider referencing the national guidelines that established these categories.
- Ethical concerns:
The manuscript does not currently mention whether ethics approval was obtained for the use of data from the newborn screening program or the selective screening cohort. It is strongly recommended that the authors include a statement in the Methods section clarifying:
- Whether ethics approval was granted;
- Whether parental informed consent was required or waived;
- How data confidentiality and privacy were ensured;
- If the NBS program includes access to follow-up genetic counselling and treatment in an equitable and ethical manner.
Author Response
Comment: [Study Aim: The aim of the study should be explicitly stated, preferably at the end of the Introduction section.]
Response: [Dear Reviewer 1, thank you so much for your edits. All edits are highlighted in red. Good afternoon, dear reviewer. The purpose of the study is formulated unambiguously and presented in a separate section according to the rules of the journal.]
Comment: [NBS Policy Background:Consider adding a dedicated paragraph in the Introduction or Discussion summarising the expansion of Russia's NBS program in 2023 (to include SMA and other conditions) and explaining how SMA was added to the national panel.]
Response: [ In the section introduction and materials and methods, information has been added on the progress of neonatal screening in the Russian Federation and diseases involving the extension.]
Comment: [Discuss the public health significance of early SMA detection within the Russian healthcare context.]
Response: [At the end of the discussion section, the importance of early detection of SMA is added.]
Comment: [Terminology – Selective vs. Universal Screening: The phrase “selective screening during NBS” (e.g., lines 350, 353) may be misleading, as NBS is defined as universal screening shortly after birth. Please clarify whether these patients were truly part of the NBS program or were referred for testing due to symptoms or other risk factors. You may wish to add a clear definition of “selective screening” vs. “universal NBS” early in the manuscript (e.g., Methods or Introduction).]
Response: [Dear reviewer, the introduction section describes the procedure for the establishment of screening in the Russian Federation, and the materials and methods section describes in detail the formation of neonatal and selective screening groups. Neonatal screening has been performed for all newborns throughout the Russian Federation since 2023. Selective screening is performed only for symptomatic patients at the direction of a clinical doctor. Selective screening during neonatal screening, this means that neonatal screening has already been started, and adult symptomatic patients are referred for selective screening. In the materials and methods section, 1 sentence has been corrected with the amendment that selective screening groups are groups of symptomatic patients.]
Comment: [Reference to National Policy and Order 274n: When referring to Order 274n (line 371), briefly explain its relevance for confirmatory testing and follow-up. It may be helpful to cite the document and explain how it mandates centralised procedures. Center Classification (3A and 3B): Clarify the roles of reference centers categorised as 3A and 3B (lines 364–368). Define their responsibilities in the NBS-confirmation-treatment workflow and consider referencing the national guidelines that established these categories.]
Response: [Dear reviewer, the materials and methods section has been supplemented with information on the structure of neonatal screening in accordance with the order. The global purpose of the program, the course of collecting biological material, and the separate role of centers 3a and 3b responsible for therapy are described. The national leadership is in the process of updating, and all recommendations are being implemented in accordance with the order.]
Comment: [Ethical concerns: The manuscript does not currently mention whether ethics approval was obtained for the use of data from the newborn screening program or the selective screening cohort. It is strongly recommended that the authors include a statement in the Methods section clarifying: Whether ethics approval was granted; Whether parental informed consent was required or waived; How data confidentiality and privacy were ensured; If the NBS program includes access to follow-up genetic counselling and treatment in an equitable and ethical manner.]
Response: [Dear reviewer, the article has a separate column about ethics at the end of the manuscript. Highlighted it in red. All ethical consents have been received and sent to the journal in a separate letter.]
Round 2
Reviewer 2 Report
Comments and Suggestions for Authors
Thank you very much to the authors for the review of the article and for the responses to the questions raised. I only have two minor suggestions to make. Figure 5 is not referenced in the text. I suggest placing it at the end of the paragraph prior to Figure 5, on the previous page of the article, page 11, line 370. In the paragraph following Figure 5, where it says 'Since January 1, 2023, newborn screening for 5q SMA has been conducted in all 89 federal subjects of the Russian Federation for congenital metabolism disorders, primary immunodeficiencies, and 5q spinal muscular atrophy.', I suggest writing 'Since January 1, 2023, newborn screening has been conducted in all 89 federal subjects of the Russian Federation for congenital metabolism disorders, primary immunodeficiencies, and 5q SMA.'
Author Response
Dear reviewer, thank you so much for noting the inaccuracy. All the edits have been taken into account and made. Added a link, changed the offer.
